# Off-season RSV epidemics in Australia after easing of COVID-19 restrictions

John-Sebastian Eden [1,2,20], Chisha Sikazwe[3,4,20], Ruopeng Xie[5,6,20], Yi-Mo Deng[7,8], Sheena G. Sullivan [7,9], Alice Michie[4], Avram Levy[3], Elena Cutmore[1,2], Christopher C. Blyth[3,10,11,12], Philip N. Britton[2,13], Nigel Crawford[14,15,16], Xiaomin Dong[7,8], Dominic E. Dwyer [2,17], Kimberly M. Edwards [5,6], Bethany A. Horsburgh[1,2], David Foley [3], Karina Kennedy[18], Cara Minney-Smith[3], David Speers[3,10], Rachel L. Tulloch[1,2], Edward C. Holmes [2], Vijaykrishna Dhanasekaran [5,6,21✉], David W. Smith [3,10,21✉], Jen Kok [17,21✉], Ian G. Barr [7,8,21✉] & the Australian RSV study group*

Human respiratory syncytial virus (RSV) is an important cause of acute respiratory infection with the most severe disease in the young and elderly. Non-pharmaceutical interventions and travel restrictions for controlling COVID-19 have impacted the circulation of most respiratory viruses including RSV globally, particularly in Australia, where during 2020 the normal winter epidemics were notably absent. However, in late 2020, unprecedented widespread RSV outbreaks occurred, beginning in spring, and extending into summer across two widely separated regions of the Australian continent, New South Wales (NSW) and Australian Capital Territory (ACT) in the east, and Western Australia. Through genomic sequencing we reveal a major reduction in RSV genetic diversity following COVID-19 emergence with two genetically distinct RSV-A clades circulating cryptically, likely localised for several months prior to an epidemic surge in cases upon relaxation of COVID-19 control measures. The NSW/ACT clade subsequently spread to the neighbouring state of Victoria and to cause extensive outbreaks and hospitalisations in early 2021. These findings highlight the need for continued surveillance and sequencing of RSV and other respiratory viruses during and after the COVID-19 pandemic, as mitigation measures may disrupt seasonal patterns, causing larger or more severe outbreaks.

A full list of author affiliations appears at the end of the paper.

Each year respiratory syncytial virus (RSV) causes an estimated 3.2 million hospital admissions and 118,200 deaths in children under five years of age, predominantly in low- and middle-income countries[1]. While this burden is greatest in the young, RSV is clinically significant for all age groups, as re-infection can occur throughout life[2]. The elderly and immuno-compromised are particularly at risk of severe infection with intensive care admission and mortality rates similar to that of influenza, posing a considerable threat to residents of long-term care facilities[3,4]. RSV causes seasonal epidemics in both tropical and temperate regions of the world[5]. In Australia, most temperate regions experience seasonal RSV outbreaks during the autumn and winter, often peaking in June–July[6] and usually preceding the influenza season[7]. In the more tropical northern parts of Australia, RSV activity correlates with the rainfall and humidity patterns of the rainy season from December to March[8].

Non-pharmaceutical interventions (NPIs) to limit the spread of severe acute respiratory syndrome coronavirus 2 (SARS-CoV-2) have disrupted the typical seasonality of other common respiratory pathogens in many countries[9,10]. Australia's initial SARS-CoV-2 epidemic was effectively controlled by NPIs[11], and those same restrictions also suppressed seasonal respiratory virus circulation, particularly for influenza virus and RSV, as the usual winter epidemics were notably absent during 2020[12–14]. While the control measures of each Australian state and territory varied in stringency and duration[15], all occurred throughout the usual peak of RSV seasonal activity. Interestingly, the impact of NPIs was not consistent across all the common respiratory viruses: rhinoviruses and, to a lesser extent, adenoviruses continued to circulate in Australia during this pandemic period after an initial disruption to usual circulation[14]. The suppression of influenza virus and RSV activity during the COVID-19 pandemic in the southern hemisphere was also seen in South Africa[16] and New Zealand[17], where similarly following an initial reduction in circulation, RSV activity rebounded in late 2020[18] and early 2021, respectively. Marked reductions in RSV activity have also been seen in the northern hemisphere since early 2020, although some European countries such as France[19], Iceland[20], as well as in Israel[21], and some US states[22] have recently reported out-of-season spikes in RSV activity in early-mid 2021. There has also been a dearth of RSV sequences submitted to public databases since early 2020, including these most recent outbreaks presumably a reflection of the lack of RSV circulation in many countries and a focus on SARS-CoV-2 sequencing.

Here, we show a major shift in the epidemiology of RSV in Australia following the emergence of SARS-CoV-2 with large scale outbreaks of disease occurring out-of-season during the summer of 2020–21. These outbreaks across three states coincided with the easing of COVID-19-related restrictions and the emergence of two novel yet related RSV-A ON1-like lineages. Using genome sequencing, we demonstrate a remarkable collapse in circulating RSV genetic diversity and uncertainty around the origins and timings of the outbreak strains due to limited global sampling. This work highlights the shifting landscape of respiratory virus molecular epidemiology in the post-COVID-19 world and the rapid pace at which activity can rebound once public health restrictions have eased. Furthermore, this work underscores the urgent need for on-going surveillance for RSV, influenza virus and other major respiratory pathogens to examine changes in their genetic diversity, particularly towards informing vaccine compositions.

## Results and discussion
### Summer outbreaks of RSV after COVID-19 restriction easing.
In late 2020, severe out-of-season RSV outbreaks occurred in several Australian states and territories beginning in New South Wales and the Australian Capital Territory (NSW/ACT) and Western Australia (WA). This was followed by outbreaks in Victoria (VIC) throughout the summer in early 2021. To understand the change in seasonal prevalence of RSV in Australia we examined RSV testing data from January 2017 to March 2021, comparing the proportion positive and overall testing capacity in NSW, WA, and VIC (Fig. 1 and Supplementary Figs. 1 and 2). Before 2020, RSV activity consistently began during mid-autumn (April–May) and normally persisted for six months with an epidemic peak in the middle of the Australian winter (middle of July, weeks 27–29). In contrast, RSV activity in 2020 occurred between six and nine months later than historically observed, and at the peak of RSV activity across each state and territory, laboratory-confirmed RSV positivity rates were considerably higher than those of the previous three seasons (Fig. 1 and Supplementary Fig. 2).

In Australia, the suppression of RSV activity in early 2020 coincided with restrictions in response to increasing community cases of COVID-19 (Fig. 1). During March 2020, individual Australian state and territory governments implemented a range of NPIs, which included limits on international arrivals and strict quarantine requirements for persons with COVID-19 (minimum of 14 days), internal border closures, social distancing, school closures or encouraging parents to keep their children home, and hygiene protocols to minimise SARS-CoV-2 transmission. Importantly, and most relevant for RSV, childcare centres mostly remained open during these restriction periods. Indeed, during the 2020/21 RSV season in Europe, where RSV activity was overall very low, the only countries with major RSV outbreaks were those with policies to keep primary school and childcare centres open throughout lockdowns[20]. Prior to the implementation of COVID-19 control measures, a gradual increase in RSV activity was observed in early 2020 across all four Australian states and territories with laboratory test RSV positivity rates being comparable to monthly averages over the previous three seasons (Supplementary Figs. 1 and 2). Introduction of NPIs led to a rapid decline in RSV incidence in each state. Examination of sentinel hospital records for bronchiolitis by ICD-10 Australian modification (AM) codes (including both RSV-confirmed bronchiolitis and bronchiolitis of unknown cause) showed a marked decline that mirrored the decrease in laboratory-confirmed RSV (Supplementary Fig. 1). The subsequent RSV epidemics in late 2020 and early 2021 resulted in test positivity and ICD-10 AM admission levels equivalent to or exceeding the normal winter seasonal RSV activity seen in any of the previous three years. For NSW, the epidemic began in September 2020 with bimodal peaks in activity in mid-November (reaching 26% positive) and early January (reaching 24% positive) (Fig. 1). The dual peaks likely reflect inconsistent testing over the Christmas and New Year holiday period, as the peak in bronchiolitis hospitalisations in NSW occurred between late December and early January and coincided with this period (Fig. 1). Furthermore, a December 2020 peak was also observed in the ACT, which is a territory within the state of NSW, and where 46% of tests were positive in this period (Supplementary Fig 2). For WA, the RSV epidemic began in late September 2020 and peaked in December 2020 at 37% positivity, with a matching peak in bronchiolitis hospitalisations (Fig. 1 and Supplementary Fig. 1).

Over the course of the pandemic, the stringency of COVID-19 restrictions has varied across the different states and territories. In VIC, a second wave of COVID-19 from July to August 2020 necessitated a longer SARS-CoV-2 control period, which likely contributed to a three-month delay in the onset of RSV activity relative to epidemic outbreaks in NSW/ACT and WA. RSV activity in VIC began in early January 2021 and then peaked in

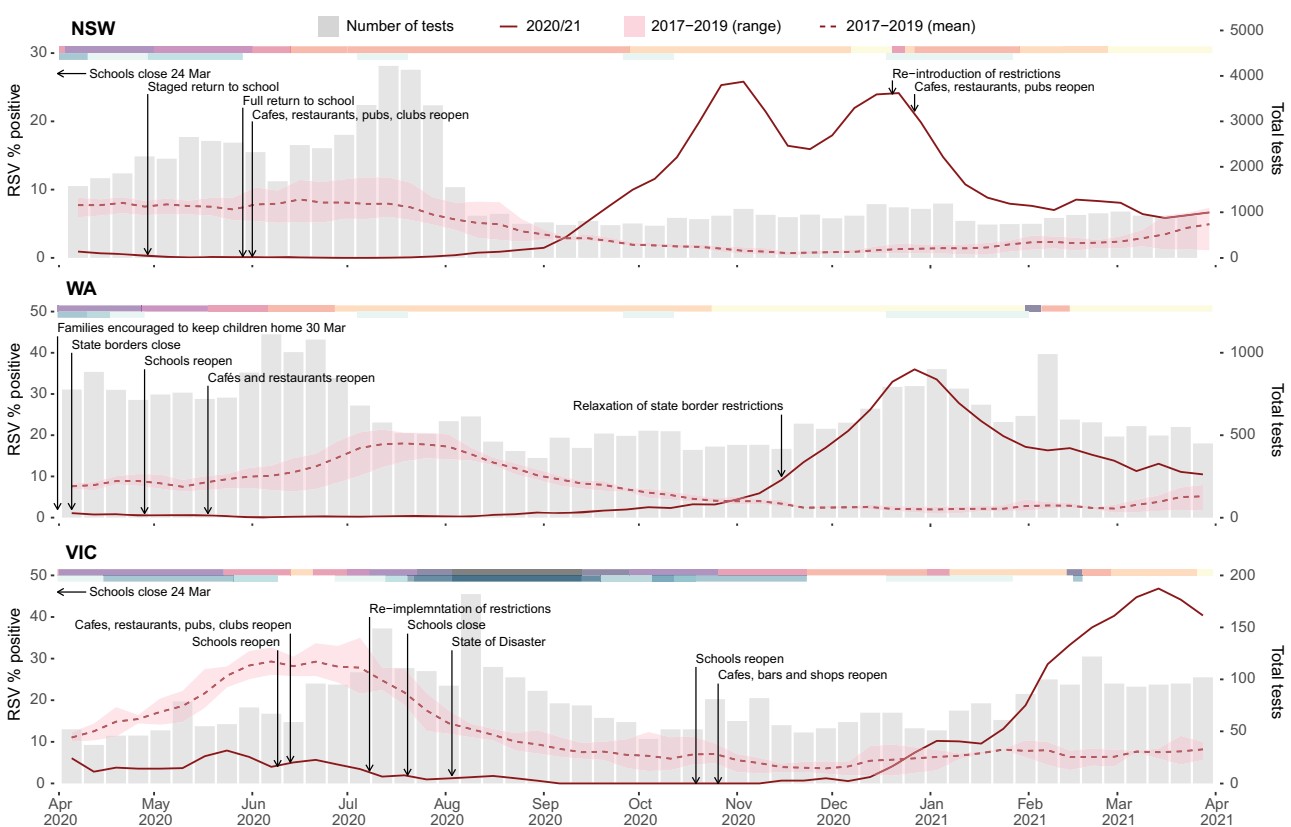

**Fig. 1 The epidemiology of RSV detections in three Australian states—New South Wales (NSW), Western Australia (WA), and Victoria (VIC).**
Laboratory testing for RSV in 2020 as weekly percent positive (red line, left y-axis) and as total number of tests performed (grey bars, right y-axis). In each panel, the dashed red line represents mean monthly RSV percent positive over the three previous seasons, and corresponding red shading represents minimum and maximum weekly percent positive. Pink-shaded bars across the top of each plot indicate the severity of pandemic restrictions, with darker colours indicative of greater stringency. Blue bars across the top of each plot indicate the periods during which students did not attend school either due to pandemic restrictions or school holiday periods, with darker colours indicative of more stringent school restrictions.

early March that year with 48% of tests positive (Fig. 1). In each state, the peaks in RSV activity occurred a few months after the relaxation of COVID-19 restrictions (Fig. 1).

**Reduced genomic diversity in the post-COVID-19 period.** Whole genome sequencing (WGS) was performed on RSV positive specimens collected before (July 2017–March 2020), during and after (April 2020–March 2021) the implementation of COVID-19 restrictions in NSW ($n = 253$), ACT ($n = 47$), WA ($n = 216$) and VIC ($n = 178$). The samples were mostly collected from young children (median ages between 0.78 and 2.34 years for the different states), although all age groups including adults and the elderly were represented. Sampling of gender was even and included geographically diverse locations (Supplementary Figs. 3 and 4). Historically in Australia, both RSV-A and -B subtypes have co-circulated with variable but relatively even prevalence[23,24]. This trend continued in the pre-COVID-19 period, where RSV-A represented 45–79% of cases. However, from late 2020 to early 2021 there was an overwhelming predominance of the RSV-A subtype (>95% for all four states and territories). This suggested that RSV-A viruses were responsible for both the NSW/ACT and WA outbreaks in late 2020, as well as the surge in RSV activity in VIC seen in early 2021.

Phylogenetic analysis of all available RSV-A genomes revealed that the Australian RSV-A viruses belonged to the ON1-like genotype first reported in Canada in December 2010[25]. These viruses have since become globally dominant and have frequently been re-introduced into Australia[23,26]. Indeed, prior to the

emergence of SARS-CoV-2 and the implementation of related control measures, multiple RSV-A ON1-like sub-lineages co-circulated (Fig. 2) with genetic diversity sustained from both endemic and imported sources[23]. While the viruses sampled before March 2020 were distributed amongst those circulating globally, RSV-A viruses circulating after the post-COVID-19 onset formed two geographically distinct monophyletic lineages (Fig. 2A). One lineage was associated with cases from NSW, ACT, and VIC, while the other was associated with cases from WA, hereafter referred to as the NSW/ACT/VIC 2020 and WA 2020 lineages, respectively (Fig. 2B). The NSW, ACT, and WA outbreaks during late 2020 occurred at a time of minimal RSV activity in other Australian states including VIC (Fig. 1 and Supplementary Fig. 1). Genomic analysis showed that cases in VIC in early 2021 were associated with multiple importations of the NSW/ACT/VIC 2020 lineage and a small number of importations of WA 2020 lineage (Fig. 2A, B).

Notably, both genetic lineages were defined by several non-synonymous changes in the genome, most of which were under significant selective pressure as measured by Contrast-FEL[27] (Fig. 2B, bold residues and Supplementary Table 2). In the WA 2020 lineage, five amino acid changes were observed under selection in the glycoprotein (T80A, T129I, and S174N), nucleocapsid (I104F) and small hydrophobic (H51Q) proteins, while 11 such lineage defining changes were identified in the NSW/ACT/VIC 2020 virus glycoprotein localised to the C-terminus region (V225A, E263Q, L265P, Y273H, S277P, S291P, Y297H, and L316P), large polymerase protein (D755G)

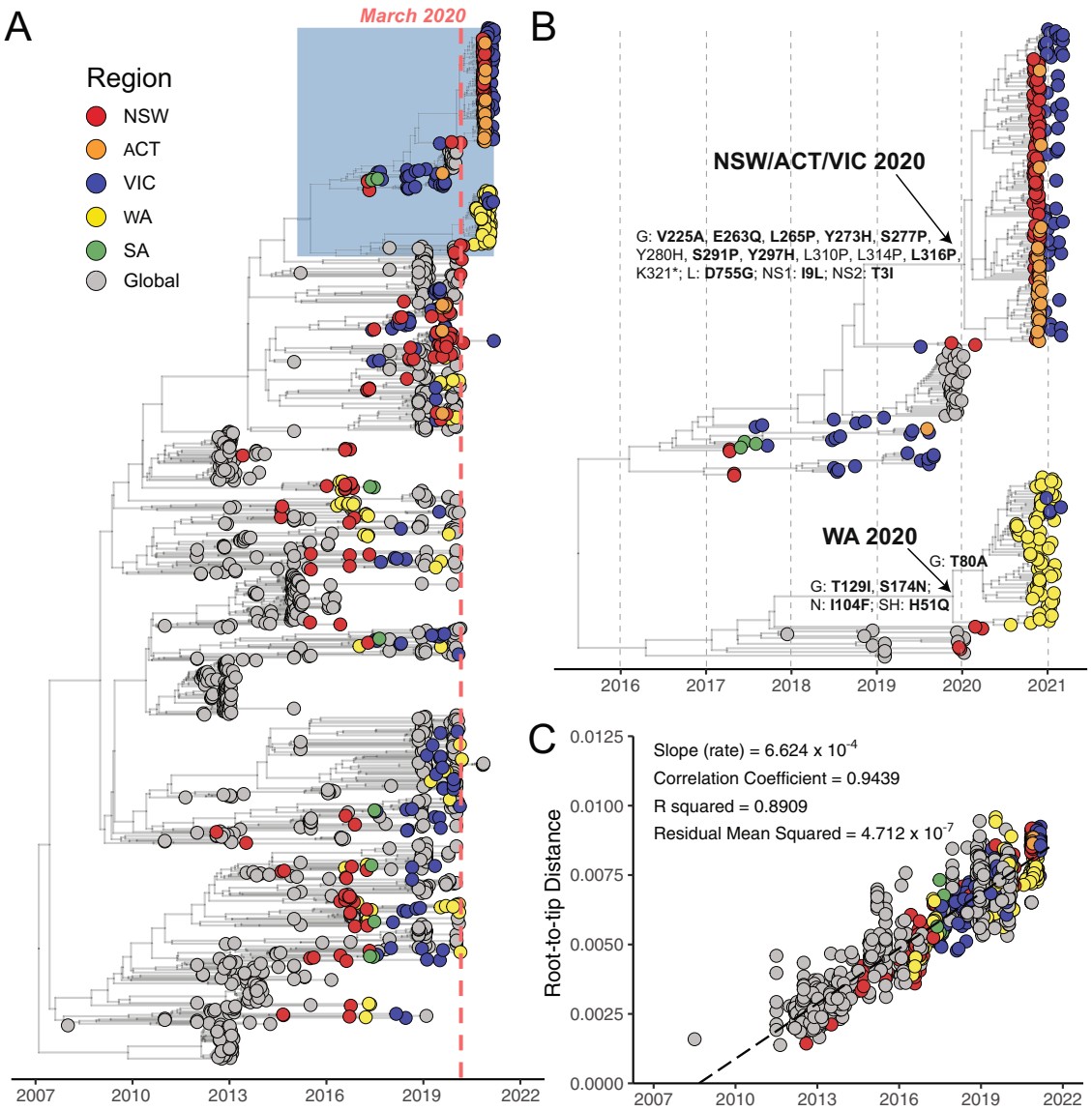

**Fig. 2 Phylogenetic analysis of global and Australian RSV-A genome sequences. A** RSV-A genome sequences were aligned with NCBI GenBank reference sequences and analysed using a Bayesian molecular clock approach estimated with BEAST (v1.10) focused on recent ON1-like viruses. Australian states—New South Wales (NSW), Australia Capital Territory (ACT), Western Australia (WA) and Victoria (VIC), and South Australia (SA)—and globally-derived sequences are coloured according to the key provided. The light red dotted line marks March 2020 and the beginning of extensive COVID-19-related restrictions. The blue shaded box is expanded in panel (**B**), which is a focused analysis of NSW/ACT/VIC and WA 2020 lineages. Amino acid mutations inferred by TreeTime are labelled on select branches, and those under significant selection pressure are shown in bold. **C** Temporal signal in RSV-A genomic dataset determined by linear-regression of root-to-tip distance (y-axis) against sample collection date (x-axis).

and non-structural proteins (NS1: I9L, NS2: T3I), most of which do not appear to have been previously reported. Despite this, an analysis using aBSREL[28], a model that tests phylogenetic branches for episodic diversifying selection, showed that branch-wise selection of the two outbreak lineages was not significant in comparison to a subset of global background sequences.

We then used RELAX[29] to further test the strength of natural selection along the outbreak branches, and found selection intensifying on NSW/ACT/VIC 2020 and WA 2020 branches (intensity parameter $k > 2$; Supplementary Table 3) in comparison to the global RSV background (i.e., sites under moderate purifying selection in the outbreak branches were subject to stronger purifying selection than non-outbreak branches). Approximately 0.25% of sites in the WA 2020 clade were under positive selection, whereas the NSW/ACT/VIC 2020 clade

contained more than 10 times as many sites (2.60–7.88%) under positive selection (Supplementary Table 3).

**Cryptic origins of the novel RSV-A lineages linked to outbreaks**. To maximise spatiotemporal sampling of the RSV-A ON1-like viruses, we expanded our phylogenetic analysis to include all available RSV glycoprotein (G) sequences (Fig. 3A), which are more numerous than published whole RSV genomes. The genome-based phylogeny contained 1520 RSV-A ON1-like genomes (Fig. 2A), and an additional 3527 sequences were included in the G gene phylogenetic analysis (Fig. 3). Despite the additional sequences, the G gene phylogeny found that the viruses from the Australian 2020–21 epidemics did not cluster with any other RSV-A viruses sampled nationally or internationally to date. As such, the initial source of these two novel RSV-A lineages

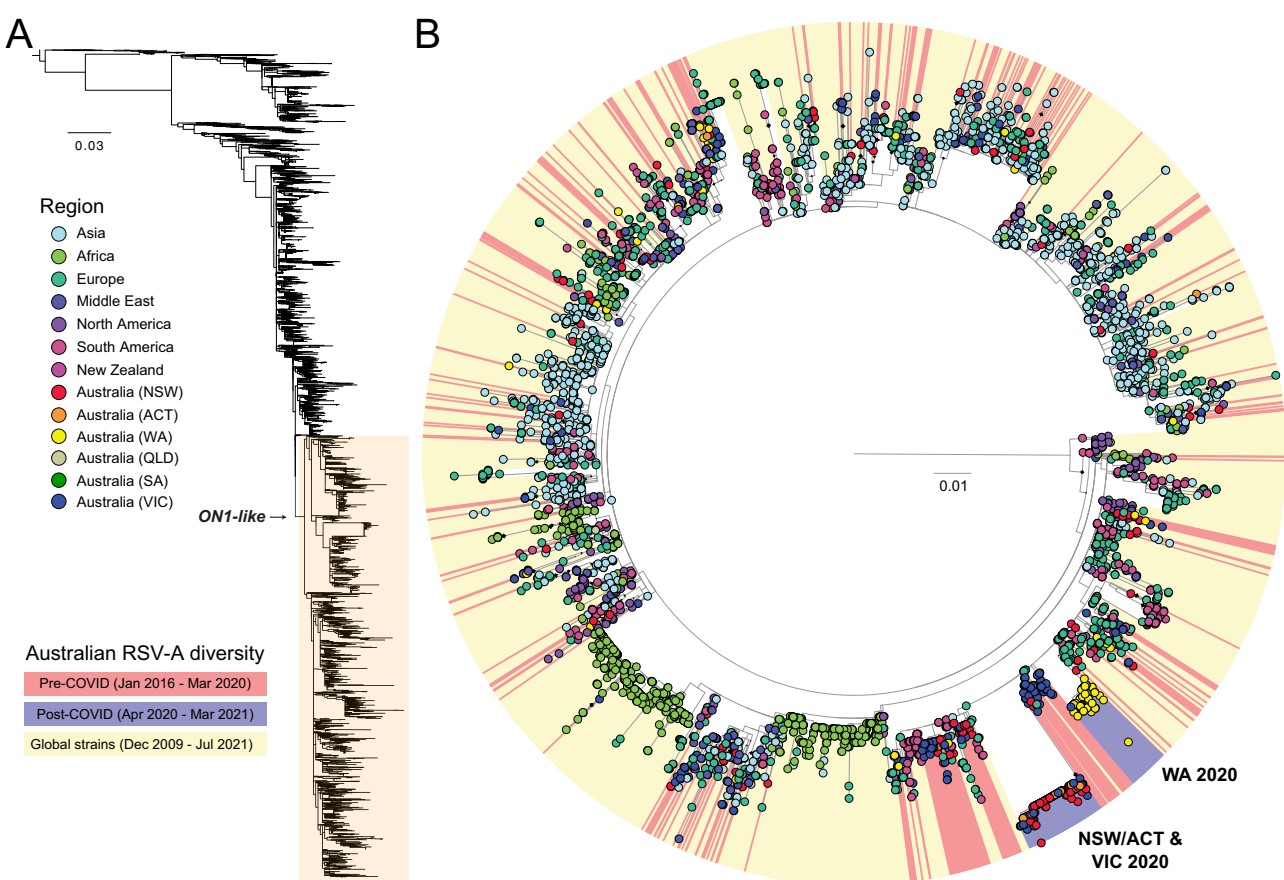

**Fig. 3 Phylogenetic analysis of global and Australian RSV-A glycoprotein sequences. A** RSV-A sequences in this study were aligned with all available RSV-A sequences from NCBI GenBank. The glycoprotein coding region was extracted, and sequences less than 300 nt were removed. **B** A detailed examination of recently circulating ON1-like viruses showed only two pre-COVID-19 lineages (coloured red) survived into the post-COVID-19 period (blue). These two lineages were associated with outbreaks in NSW/ACT and WA in late 2020, and VIC in early 2021. No sequences sourced globally (yellow) were found to be related to the these lineages, suggesting the sources remain unknown. Australian states—New South Wales (NSW), Australia Capital Territory (ACT), Western Australia (WA) and Victoria (VIC), South Australia (SA) and Queensland (QLD)—and globally-derived sequences are coloured according to the key provided. Diamonds at nodes indicate bootstrap support values >70%. Branches are proportional to the number of nucleotide substitutions per site.

remains undetermined (Fig. 3A, B). This most likely reflects a lack of RSV genomic surveillance during the COVID-19 pandemic, where clear gaps remain in sampling (Supplementary Fig. 5). However, like the genome-scale analysis, the G gene phylogeny revealed a major genetic bottleneck through which the genetic diversity of other ON1-like lineages had co-circulated from 2016 and 2020 were largely eliminated during these outbreak periods (Fig. 3B). We also examined RSV-B diversity, and while the number of detections were very low in the post-COVID-19 period, a similar pattern was observed to RSV-A diversity, whereby previously established lineages were mostly absent, and a single lineage was dominant during the 2020–21 outbreak (Supplementary Fig 6). Taken together, these results illustrate a remarkable collapse in the genetic diversity of RSV in Australia during the implementation of COVID-19 related restrictions (Fig. 3B).

Phylogenetic analysis showed there was sufficient genetic diversity within the outbreak samples to indicate the circulation of these viruses in NSW and WA prior to implementation of COVID-19 restrictions (Fig. 2B). The NSW/ACT/VIC 2020 and WA 2020 genomes contained extensive rate variation ($R^2 < 0.2$) and did not conform to temporality in a root-to-tip regression (Supplementary Fig. 7), which is essential for accurate estimation of divergence times[30]. Hence, we inferred the time to most recent

common ancestor (tMRCA) of NSW/ACT/VIC and WA 2020 lineages from the large Bayesian molecular clock analysis presented in Fig. 2A, B, followed by a post hoc correction for excess deleterious mutations[31] and/or selection pressure (Supplementary Tables 3 and 4), as described previously[32]. The corrected tMRCA estimates suggest origins around March 2020 (Table 1), with confidence intervals spanning the period immediately prior to implementation of pandemic restrictions in Australia and globally (Fig. 1). Surveillance data support this inference for the WA 2020 lineage, which was first detected in central and southern non-metropolitan regional WA from late July to September, then in the Perth metropolitan area in October (Supplementary Fig. 4), indicating low-level regional circulation and rapid spread soon after introduction into areas of high population density, in this case, metropolitan Perth. The genomic data also showed two transcontinental transmissions from WA to VIC after November 2020, at a time when interstate travel bans had eased (Figs. 2 and 3). In contrast, the origin and early dissemination of the NSW/ACT/VIC 2020 lineage was less certain.

Despite the widespread occurrence and relatively high genetic diversity of outbreaks in NSW and ACT (Figs. 2 and 3B), no precursor virus(es) of the same lineage were detected prior to the outbreak onset, despite efforts to identify and sequence cases from the low-activity period in early-mid 2020. In addition to a

**Table 1 Mean time of most recent common ancestors (tMRCA) for NSW/ACT/VIC 2020 and WA 2020 clades estimated by different methods.**

| Method | tMRCA (NSW/ACT/VIC)[a] | tMRCA (WA)[a] | Evolution rate (per site per year) |
|---|---|---|---|
| Beast | 2020-01-10 [2019-11-05, 2020-03-16] | 2019-11-23 [2019-08-08, 2020-02-20] | $6.624 \times 10^{-4}$ |
| Beast-SLAC[b] | 2020-05-03 [2020-03-16, 2020-06-20] | 2020-06-03 [2020-04-01, 2020-07-25] | NSW/ACT/VIC: $9.092 \times 10^{-4}$ WA: $1.124 \times 10^{-3}$ |
| Beast-SNAP[b] | 2020-03-25 [2020-01-31, 2020-05-18] | 2020-05-06 [2020-02-26, 2020-07-03] | NSW/ACT/VIC: $8.070 \times 10^{-4}$ WA: $1.021 \times 10^{-3}$ |
| Beast-Contrast-FEL[b] | 2020-05-08 [2020-03-22, 2020-06-25] | 2020-06-01 [2020-03-30, 2020-07-24] | NSW/ACT/VIC: $9.245 \times 10^{-4}$ WA: $1.117 \times 10^{-3}$ |

[a]Ranges in brackets show confidence intervals (95% highest probability densities).
[b]tMRCA corrected according to *dN/dS* ratio.

lack of RSV positive samples from the middle of 2020, our analysis was hampered by an inflated evolutionary rate due to recent sampling[33], multiple novel non-synonymous substitutions in the G gene, and variable evolutionary rates which complicated reliable phylogenetic dating estimates. As per our estimates of the origins for NSW/ACT/VIC 2020 and WA 2020 lineages, the viruses were already likely to be present in the country during the early stages of the COVID-19 pandemic. However, the origins of these novel virus lineages into Australia remains unclear as the branches leading to the outbreaks represent an unsampled diversity of >1 year for NSW/ACT/VIC and >2 years for WA lineage prior to March 2020 (Table 1), compounded by major biases in recent RSV genomic surveillance (Supplementary Fig. 5).

An examination of RSV diversity in Australia before and after the implementation of COVID-related NPIs using WGS has shown a major collapse in lineages that circulated prior to 2020[23,26]. This coincided with the emergence of two distinct, but phylogenetically related RSV-A lineages associated with summer outbreaks in NSW/ACT and WA, respectively. Both lineages were subsequently imported into VIC, where the NSW/ACT lineage caused a major outbreak in early 2021. Our analysis suggests cryptic circulation of both of these lineages, while other RSV-A and -B lineages were largely eliminated through COVID-19 related NPIs. While the estimation of the source of the epidemic was hindered by a paucity of sequence data from other jurisdictions and globally, the genetic diversity observed during these outbreaks strongly indicates undetected local circulation prior to the widespread outbreak. Travel restrictions and other social distancing measures may have significantly reduced RSV transmission and slowed spread but were unable to eliminate RSV in metropolitan WA and in NSW, and after a considerable delay, a substantial out-of-season epidemic occurred.

The near absence and subsequent resurgence of RSV-A in Australia has provided a unique opportunity to increase our understanding about how RSV epidemics occur and to identify measures for better control of RSV and other respiratory viruses in the future. The outbreak branches were characterised by intensifying selection, and a distinct set of mutations with unsampled ancestral branches extending over 1 and 2 years, respectively. This indicates these were distinct genotypes that had evolved under different evolutionary constraints in comparison to the general RSV background; the reasons for which require further study. Furthermore, our study highlights how quickly respiratory pathogens can rebound, even leading to unseasonal epidemics. Delayed or forgone RSV seasons may increase the cohort of young children susceptible to RSV infection and increase the age of first infection leading to larger outbreaks of RSV when they do finally occur. Increasing the age of first

infection may be expected to coincide with reduced hospitalisations given that RSV burden is most pronounced in infants less than 6 months old[34]; however, this has not been reflected in bronchiolitis admissions (Supplementary Fig 1), with peak admissions in WA and VIC higher than in prior seasons. By increasing the pool of susceptible children, including those with underlying risk factors such as congenital heart disease, extreme prematurity, or chronic lung disease, outbreaks may also be more severe with regard to hospitalisations and intensive care admissions. Recent modelling studies predict delayed and severe RSV outbreaks in the US during the 2021–2022 winter[9] but not early out-of-season outbreaks as was observed for some states[35]. Whether these large summer epidemics experienced in NSW/ACT, WA, and VIC have sufficiently reduced the susceptible population to result in a smaller than usual winter seasons going forward is uncertain. The early winter data of 2021 appears to support this premise in WA and VIC and possibly to a lesser extent in NSW/ACT.

It also remains unclear how long it will take for normal winter RSV seasonality to resume in Australia and globally. The H1N1 2009 influenza pandemic impacted respiratory virus circulation for a number of years[36]. The findings from this study exemplify the need to be prepared for the occurrence of large outbreaks of RSV outside of normal seasonal periods and for health systems to be prepared to combat future severe RSV outbreaks. It also raises important questions as to how the epidemiological and evolutionary dynamics of RSV outbreaks might inform the re-emergence of influenza virus, which is still expected, and given the smaller role children play in population-scale transmission of influenza compared to RSV, may require the re-opening of international borders to import the influenza variants required to effectively seed new local outbreaks. Nonetheless, our study highlights the propensity for COVID-19-related NPIs to cause immense disruption in seasonal patterns of respiratory virus circulation and evolution. Furthermore, this study provides a timely warning to countries emerging from pandemic restrictions: the burden of disease from other respiratory pathogens such as RSV, may have all but disappeared and will likely rebound in the near future, possibly at unusual times and with stronger impact.

## Methods

**RSV surveillance and epidemiology**. Respiratory specimen testing with quantitative RT-PCR assays was performed at six sites including (i) NSW Health Pathology—Institute of Clinical Pathology and Microbiology Research (ICPMR), Westmead, NSW, (ii) The Children's Hospital at Westmead, NSW, (iii) PathWest Laboratory Medicine WA, Perth, WA[37], (iv) The Royal Children's Hospital, Melbourne, VIC, (v) Monash Pathology, Monash Medical Centre, Clayton, VIC and (vi) ACT Pathology, Canberra, ACT. NSW Health Pathology—ICPMR and PathWest laboratories are both major diagnostic hubs that provide state-wide testing for respiratory viruses in NSW and WA, respectively. Monash Pathology provides services for all ages for a region of Melbourne, while the Children's

Hospital at Westmead and Royal Children's Hospitals are major metropolitan hospitals in NSW and VIC, respectively. ACT Pathology provides diagnostic services for all adult and paediatric hospital emergency department presentations and a proportion of outpatient community requests for the ACT. Weekly counts for RSV testing were collated for the period January 2017–March 2021, and derived from three laboratories: PathWest Laboratory in Perth, WA, NSW Health Pathology—ICPMR in Sydney, NSW, ACT Pathology, Canberra, ACT, and the Bio21 Royal Children's Hospital in Melbourne, VIC. PathWest, ICPMR, and ACT Pathology are public health laboratories testing children and adults across their respective states, whereas Bio21 only provided testing data for children receiving care at the Royal Children's Hospital in Melbourne, and therefore only includes results for children under 18 years of age. The proportion of tests that were RSV positive was calculated and smoothed using a 3-week, centred moving average. Data were plotted in time series to compare observed RSV activity for April 2020–March 2021, versus the average for April 2017–March 2020. Monthly bronchiolitis admissions for the three children's hospitals in Perth, Sydney, and Melbourne were collated for the period January 2017–March 2021. Only admissions with a J21 ICD-10 AM code were considered. Prior work has shown that admissions for bronchiolitis are heavily represented by children aged <2 years. School and other restrictions for each state were collated from media releases and official public health directions issued in each of the three states. In addition, school holiday periods for all years January 2017–March 2021 were collated to visually assess the role of school holidays as a proxy for student mixing in RSV seasonality. The period between relaxation of restrictions and increased RSV activity was visually compared.

**RSV subtyping and whole genome sequencing**. Samples were sequenced from cases collected for routine diagnostic purposes as part of public health responses and from on-going research studies with approval from the local Human Research Ethics Committees of the Royal Children's Hospital and Western Sydney Local Health District with approval numbers 37185 and LNR/17/WMEAD/128, respectively. Total nucleic acid was extracted from RSV positive respiratory specimens archived at −80 °C using high-throughput bead-based protocols. RSV WGS was conducted using established protocols[24] for a subset of samples selected to provide temporal and geographical representation of (i) the pre-COVID-19 period, inclusive of July 2017–March 2020, and (ii) the post-COVID-19 period, inclusive of April 2020–March 2021. Briefly, viral cDNA was prepared from extracted nucleic acid using SuperScript IV VILO Master Mix or SuperScript IV (Invitrogen, Carlsbad, CA, USA), followed by RT-PCR amplification of four long overlapping fragments spanning the RSV genome using Platinum SuperFi Master Mix (Invitrogen). The four target amplicons were then combined equally before DNA purification with AMPure XP (Beckman Coulter, Indianapolis, IN, USA). The purified and pooled amplicons were diluted to 0.2 ng/μl and prepared for sequencing using the Nextera XT library preparation kit with v2 indexes (Illumina, San Diego, CA, USA). Multiplexed libraries were then sequenced either on an Illumina iSeq 100 or MiSeq producing at least 200,000 paired end reads (2x150nt) per library. For genome assembly, the sequence reads were QC trimmed using BBDuk v37.98[38] before de novo assembly with MEGAHIT v1.1.3[39] or reference-based assembly with IRMA[40]. To confirm assembly, the trimmed sequence reads were re-mapped onto the draft genome with BBMap v37.98 and visually assessed using the Geneious Prime v.2020.0.3 before the final majority consensus genome was extracted.

**Phylogenetic analysis**. RSV sequences generated in this study were analysed along with reference sequences sourced from NCBI GenBank and GISAID databases (see Supplementary Data 2) or from the NIAID Virus Pathogen Database and Analysis Resource (ViPR)[41] at http://www.viprbrc.org/. Specifically, all available full-length genomes and partial G gene sequences (greater than 300 nt) with collection dates were downloaded on 1st October 2021. Multiple sequence alignments were performed independently with MAFFT v.7[42] and examined using TempEst v.1.5[30] to identify and exclude excessively divergent sequences in a preliminary maximum likelihood (ML) tree generated in FastTree v.2.1[43]. G gene phylogenies were estimated using RAxML v.8[44] using the GTR-$\Gamma$ nucleotide substitution model, with branch support estimated by 1000 bootstrap replicates.

Time-scaled phylogenetic trees of the full-length alignments were inferred using a Bayesian molecular clock phylogenetic analysis pipeline proposed by du Plessis et al.[45] for large datasets using BEAST (v1.10)[46]. In this pipeline, a maximum likelihood tree with branch lengths in nucleotide substitutions (genetic distances) and a time-calibrated tree estimated using TreeTime[47] were used as starting inputs for Bayesian dating. To maintain a map between branches of both trees, any clade disruptions were rejected, and only polytomy resolutions and branch durations could be optimised under a strict clock model with a log-normal prior mean rate of $6.624 \times 10^{-4}$ substitutions/site/year, which we inferred using a root-to-tip regression in TempEst v.1.5[30] based on a ML tree generated in IQ-TREE v.2.0[48] with the best-fit nucleotide substitution model. The Skygrid population model with 14 grid points (13.18 years duration in this study) and a Laplace root-height prior with mean equal to the time-calibrated tree estimated by TreeTime was used with scale set to 20% of the mean. We performed two MCMC runs for 100 million steps, sampled every 10,000 steps, and discarded 10% as burn-in, ensuring ESS for each parameter is larger than 200 using Tracer v.1.7.1. We estimated the effect of excess

non-synonymous/synonymous substitutions in outbreak clades on the tMRCA estimates using a protocol described previously[32], in which SLAC[49] (Single Likelihood Ancestor Counting), SNAP v.2.1.1[50] (Synonymous Non-synonymous Analysis Programme) and Contrast-FEL were used to estimate the difference between the mean ratio of non-synonymous substitutions per non-synonymous site to synonymous substitutions per synonymous site ($d_N/d_S$) of the NSW/ACT/VIC ($n = 166$) and WA ($n = 79$) datasets versus a subsampled global dataset ($n = 112$) comprising genomes from major clades.

aBSREL[28] was used to test whether a proportion of sites evolved under positive selection in any of the branches, and RELAX[29] was applied to test for strength of selection (relaxed/intensified) in a set of test branches. In this study, the clades of the subsampled global dataset were defined as background, and the NSW/ACT/VIC and WA clades were regarded as foreground separately. Furthermore, Contrast-FEL[27] was used to identify codon sites that evolved differently for the outbreak clades. For this, the subsampled global dataset, four sequences from the largest NSW/ACT/VIC 2020 clade and three sequences from the largest WA 2020 clade were used, with the ancestral branches of the outbreak clades assigned as foreground for each of the two clades respectively.

**Reporting summary**. Further information on research design is available in the Nature Research Reporting Summary linked to this article.

## Data availability

The sequence data generated in this study have been deposited in the NCBI GenBank database under accession codes OM857140 - OM857397, and the GISAID EpiRSV database with accession numbers EPI_ISL_1653938 to EPI_ISL_1653948, EPI_ISL_2543762 to EPI_ISL_2543853, and EPI_ISL_2839170 to EPI_ISL_2839457 (see Supplementary Table 1 and Supplementary Data 1). Reference sequences were downloaded from the NCBI GenBank and GISAID EpiRSV databases (see Supplementary Data 2).

## Code availability

The code used for genome assembly can be found at https://github.com/jsede/virus_assembly/ (https://doi.org/10.5281/zenodo.6360724)[51], while code for individual analyses can be found at https://github.com/jsede/RSV_2020 (https://doi.org/10.5281/zenodo.6360678)[52].

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

## Acknowledgements
The WHO Collaborating Centre for Reference and Research on Influenza is supported by the Australian Government Department of Health. Funding was provided through the ICPMR Private Practice Trust fund to J.S.E., and J.K., the National Health and Medical Research Council Centre of Research Excellence in Emerging Infectious Diseases (#1102962) to J.S.E., E.C.H., and D.S.W. and the Sydney Institute for Infectious Diseases at the University of Sydney to J.S.E. We would finally like to thank all the authors who have kindly shared genome data on GISAID to the EpiRSV database (Supplementary Data 3).

## Author contributions
This study was designed by J.S.E., C.S., S.S., V.D., D.W.S., J.K., and I.G.B. Epidemiological data and sample provision were performed by S.S., A.L., C.C.B., P.N.B., N.C., D.E.D, K.M.E., D.F., K.K., D.S., D.W.S., and J.K. Genome sequencing was performed by J.S.E., C.S., Y.M.D., A.M., E.C., X.D., B.A.H., C.M.S., and R.L.T. Phylodynamic analysis was performed by J.S.E., C.S., R.X., X.D., E.C.H., and V.D. Data visualisation was performed by J.S.E., C.S., R.X., S.S., and V.D. Project supervision was performed by E.C.H., V.D., D.W.S., J.K., and I.G.B. The original draft of this manuscript was prepared by J.S.E., C.S., R.X., S.S., V.D., D.W.S., J.K., and I.G.B. and was reviewed and edited by all remaining authors.

## Competing interests
The authors declare no competing interests.

## Additional information

[1]Centre for Virus Research, Westmead Institute for Medical Research, Westmead, NSW 2145, Australia. [2]Sydney Institute for Infectious Diseases, Sydney Medical School, The University of Sydney, Sydney, NSW 2006, Australia. [3]PathWest Laboratory Medicine WA, Department of Microbiology, Nedlands, WA 6009, Australia. [4]School of Biomedical Sciences, The University of Western Australia, Crawley, WA 6009, Australia. [5]School of Public Health, LKS Faculty of Medicine, The University of Hong Kong, Hong Kong, China. [6]HKU-Pasteur Research Pole, School of Public Health, LKS Faculty of Medicine, The University of Hong Kong, Hong Kong, China. [7]WHO Collaborating Centre for Reference and Research on Influenza, Royal Melbourne Hospital, at the Peter Doherty Institute for Infection and Immunity, Melbourne, VIC 3000, Australia. [8]Department of Microbiology and Immunology, University of Melbourne, at the Peter Doherty Institute for Infection and Immunity, Melbourne, VIC 3000, Australia. [9]Department of Infectious Diseases, University of Melbourne, at the Peter Doherty Institute for Infection and Immunity, Melbourne, VIC 3000, Australia. [10]School of Medicine, The University of Western Australia, Crawley, WA 6009, Australia. [11]Department of Infectious Diseases, Perth Children's Hospital, Nedlands, WA 6009, Australia. [12]Wesfarmers Centre of Vaccines and Infectious Diseases, Telethon Kids Institute, Nedlands, WA 6009, Australia. [13]Departments of Infectious Diseases and Microbiology, The Children's Hospital at Westmead, Westmead, NSW 2145, Australia. [14]Murdoch Children's Research Institute, Melbourne, VIC 3000, Australia. [15]Department of Paediatrics, University of Melbourne & Royal Children's Hospital, Melbourne, VIC 3000, Australia. [16]Immunisation Service, Royal Children's Hospital, Melbourne, VIC 3000, Australia. [17]NSW Health Pathology - Institute for Clinical Pathology and Medical Research, Westmead Hospital, Westmead, NSW 2145, Australia. [18]Departments of Clinical Microbiology and Infectious Diseases, Canberra Hospital, Garran, ACT 2605, Australia. [20]These authors contributed equally: John-Sebastian Eden, Chisha Sikazwe, Ruopeng Xie. [21]These authors jointly supervised this work: Vijaykrishna Dhanasekaran, David W. Smith, Jen Kok, Ian G. Barr. *A list of authors and their affiliations appears at the end of the paper. ✉email: veej@hku.hk; David.Smith@health.wa.gov.au; jen.kok@health.nsw.gov.au; ian.barr@influenzacentre.org

## the Australian RSV study group

Annette Alafaci[14], Ian Carter[17], Andrew Daley[15], Michelle Francis[19], Alison Kesson[2,13], Hannah Moore[12], Christine Ngo[17] & Tyna Tran[17]

[19]Microbiology Laboratory, Monash Pathology, Monash Health, Melbourne, VIC, Australia.

