## [Peer Review File · Nature Communications]

Off-season RSV epidemics in Australia after easing of COVID-19 restrictionsREVIEWER COMMENTS

Reviewer #1 (Remarks to the Author):

Eden and colleagues demonstrate that RSV experienced a massive population bottleneck in Australia due to COVID-19 restrictions, but that RSV experienced a non-seasonal epidemic in 2020. These findings are of great interest and importance relating to the impacts of COVID-19 on other respiratory pathogens and on RSV in its own right.

The authors highlight numerous non-synonymous and synonymous substitutions along the branches leading to the emergent clades. First, I would not call these mutations "significant", as this term evokes a statistic inference. Second, I would suggest the authors do perform a statistic test to determine if evolution along these branches is different from the rest of the tree. Two suggested methods are aBS-REL and CONTRAST-FEL, both well suited for this task. In the absence of such a formal statistical test, I worry that any emphasis placed on these mutations may be misplaced.

The large polytomies present at the base of the emergent clades in the tree in Figure 2 suggest the molecular clock method employed (LSD) is likely inadequate to date the TMRCA of these clades. I strongly suggest the authors instead employ a coalescent framework in either BEAST or TreeTime. While important from a descriptive perspective, the overall findings of this paper will remain unchanged. What may also be gained from a BEAST or TreeTime analysis is phylogeographic reconstruction, which will substantially inform the discussion of geographic origins, which is currently based on analysis at the tips of the tree, rather than internal nodes. These modifications will substantially improve the quality of this manuscript.

Minor Points:

Abstract. The two clades are not an exception to a reduction in genetic diversity, they are the evidence of it.

Figure 1. Please include the full state names in the Figure Legend, along with their abbreviation.

Reviewer #2 (Remarks to the Author):

manuscript NCOMMS-21-28673-T "Off-season RSV epidemics in Australia after easing of COVID-19 restrictions" by John-Sebastian Eden et al.

The authors have monitored RSV evolution patterns in Australia during the COVID-19 era. RSV surveillance continued and a striking pattern of RSV diversity was observed with an apparently Australian-specific lineage emerging and spreading through the country. Due to lockdown measures the typical RSV patterns have been influenced. Studying these patterns revealed important features of RSV transmission and evolution. Especially the emergence of local strains and their spread through Australia revealed the potential of RSV to adjust to new transmission conditions. The conclusions are well supported, the figures are clear and the methodology sound and well described. The authors should consider the following points.

1. Line 165: "While the viruses sampled before March 2020 were well-dispersed amongst those circulating globally, viruses from the post-COVID-19 period formed two geographically distinct monophyletic lineages (Figure 2A)."

Is it possible that global RSV genomic surveillance has not continued at the same pace, thus Australian RSV looks to be unique to Australian but this could be due to the lack of global RSV genomes reported since COVID-19 began. The authors should comment on this.

2. It would be also useful to see the number of global (non-Australian) vs Australian sequences for the different time periods for the sequence data used for Figure 2a and 3a. I suspect that global RSV surveillance and especially sequencing has not continued at same pace during the epidemic and what appears to be Australian-specific evolution could simply be due to Australian maintaining their surveillance while the rest of the world has slowed.

3. For completeness, these recent publications on RSV patterns during COVID-19 should be cited and discussed.

van Summeren J et al. Low levels of respiratory syncytial virus activity in Europe during the 2020/21 season: what can we expect in the coming summer and autumn/winter?. *Euro Surveill.* 2021;26(29):pii=2100639.
<https://doi.org/10.2807/1560-7917.ES.2021.26.29.2100639>

Olsen SJ, Winn AK, Budd AP, et al. Changes in Influenza and Other Respiratory Virus Activity During the COVID-19 Pandemic — United States, 2020–2021. *MMWR Morb Mortal Wkly Rep* 2021;70:1013–1019. DOI: <http://dx.doi.org/10.15585/mmwr.mm7029a1external icon>.

4. Line 194: "Despite the additional sequences, the G protein phylogeny found that the viruses from the 2020-2021 epidemics did not cluster with any other viruses sampled nationally or internationally up to date."

A comment on how reduced Australian and global sequencing might influence this analysis and the conclusions on the level of observed diversity.

RESPONSES TO REVIEWER COMMENTS

We thank the reviewers for finding our study important and interesting. We have addressed each of the comments below and made needed changes in the manuscript, highlighting each of the changes below.

Reviewer #1 (Remarks to the Author):

1. Eden and colleagues demonstrate that RSV experienced a massive population bottleneck in Australia due to COVID-19 restrictions, but that RSV experienced a non-seasonal epidemic in 2020. These findings are of great interest and importance relating to the impacts of COVID-19 on other respiratory pathogens and on RSV in its own right.

The authors highlight numerous non-synonymous and synonymous substitutions along the branches leading to the emergent clades. First, I would not call these mutations “significant”, as this term evokes a statistic inference. Second, I would suggest the authors do perform a statistic test to determine if evolution along these branches is different from the rest of the tree. Two suggested methods are aBS-REL and CONTRAST-FEL, both well suited for this task. In the absence of such a formal statistical test, I worry that any emphasis placed on these mutations may be misplaced.

Per reviewer’s request, we included the suggested analyses of the long branches leading to RSV epidemics using Contrast-FEL and aBSREL (see Lines 184-194). Interestingly, Contrast-FEL identified 16 positively selected sites on both major branches leading to the NSW/ACT/VIC and WA lineages, with significantly higher non-synonymous substitution rates ($p < 0.05$). These results are now summarized in Table S2. However, aBSREL showed that the branch-wise selection was not significant among the RSV lineages, except for two tips branches (data not shown). While Contrast-FEL analysis suggested higher average dN/dS ratios for the key branches relative to background (Table S4), the values were both below one, indicating mildly deleterious mutations were occurring and that many of these mutations are transient and will likely be purged by negative selection. This may explain the lack of episodic diversifying selection by aBSREL.

We conducted further hypothesis testing for strength of selection, using RELAX, which uses the aBSREL framework to estimate ω separately for the test (outbreak clades) and reference branches (background) against a null model with same ω distribution for test and reference branches. We found that both outbreak clades were under strong intensifying selection (intensity parameter $k > 2$) under multiple replicates (Table S3). The results have been discussed in Lines 196-202.

2. The large polytomies present at the base of the emergent clades in the tree in Figure 2 suggest the molecular clock method employed (LSD) is likely inadequate to date the TMRCA of these clades. I strongly suggest the authors instead employ a coalescent framework in either BEAST or TreeTime. While important from a descriptive perspective, the overall findings of this paper will remain unchanged. What may also be gained from a BEAST or TreeTime analysis is phylogeographic reconstruction, which will substantially inform the discussion of geographic origins, which is currently based on analysis at the tips of the tree, rather than internal nodes. These modifications will substantially improve the quality of this manuscript.

Per Reviewer’s request, we replaced Figure 2 with the dated tree generated using BEAST (v1.10) and have provided comment on the new results in Lines 243-260 along with a description of methods in Lines 396-415.

Because selection pressure or excess deleterious mutations in intensively sampled outbreak clades will affect tMRCA estimates, we performed a correction for excess dN/dS as described previously (Smith *et al.* <https://www.nature.com/articles/nature08182>). These results are summarised in new Tables 1 & S4. However, we interpret these findings with some degree of caution (Lines 268-279), because in addition to excess non-synonymous mutations, the outbreak datasets (NSW/ACT/VIC and WA) showed extensive rate variation and did not show temporality in a root-to-tip regression ($R^2 < 0.2$) (Figure S7). Hence, we felt we could not particularly rely on the genomic data to provide any robust phylogeographic inference, and instead reflected on the spread of the outbreak based on epidemiological data (Lines 253-257).

Minor Points:

3. Abstract. The two clades are not an exception to a reduction in genetic diversity, they are the evidence of it.

Modified as suggested.

4. Figure 1. Please include the full state names in the Figure Legend, along with their abbreviation.

Modified as suggested.

Reviewer #2 (Remarks to the Author):

The authors have monitored RSV evolution patterns in Australia during the COVID-19 era. RSV surveillance continued and a striking pattern of RSV diversity was observed with an apparently Australian-specific lineage emerging and spreading through the country. Due to lockdown measures the typical RSV patterns have been influenced. Studying these patterns revealed important features of RSV transmission and evolution. Especially the emergence of local strains and their spread through Australia revealed the potential of RSV to adjust to new transmission conditions. The conclusions are well supported, the figures are clear and the methodology sound and well described. The authors should consider the following points.

1. Line 165: "While the viruses sampled before March 2020 were well-dispersed amongst those circulating globally, viruses from the post-COVID-19 period formed two geographically distinct monophyletic lineages (Figure 2A)."

Is it possible that global RSV genomic surveillance has not continued at the same pace, thus Australian RSV looks to be unique to Australian but this could be due to the lack of global RSV genomes reported since COVID-19 began. The authors should comment on this.

Publicly available global RSV genomic surveillance has generally been very limited in recent years, with very low sequence data immediately prior and during the COVID-19 pandemic. As suggested in comment 2 below, we included a new Supplementary Figure (Figure S5) to show the number of global and Australian sequences (partial and whole) available in recent years and used in our analyses. In the revised manuscript, we also included an further 390 genome and 1,723 glycoprotein sequences that were submitted to GISAID and the NCBI GenBank database since our paper was initially submitted, which included sequences collected during 2019-2020, predominantly from Europe. Notably, this new data does not change the initial observation that the two outbreak lineages are

unique, and adds some support for potential local circulation of this clade within Australia. However, clear sampling biases are present as we have expanded the Discussion to include these limitations in Lines 276-279.

2. It would be also useful to see the number of global (non-Australian) vs Australian sequences for the different time periods for the sequence data used for Figure 2a and 3a. I suspect that global RSV surveillance and especially sequencing has not continued at same pace during the epidemic and what appears to be Australian-specific evolution could simply be due to Australian maintaining their surveillance while the rest of the world has slowed.

Please see response to comment 2 immediately above. In Figure S5, we compare sampling frequencies to show global surveillance has been low prior to the COVID-19 pandemic, and reflect on this in the manuscript, see Lines 220-222 & Lines 276-279.

3. For completeness, these recent publications on RSV patterns during COVID-19 should be cited and discussed.

van Summeren J et al. Low levels of respiratory syncytial virus activity in Europe during the 2020/21 season: what can we expect in the coming summer and autumn/winter?. *Euro Surveill.*2021;26(29):pii=2100639.

Olsen SJ, Winn AK, Budd AP, et al. Changes in Influenza and Other Respiratory Virus Activity During the COVID-19 Pandemic — United States, 2020–2021. *MMWR Morb Mortal Wkly Rep* 2021;70:1013–1019.

We thank the reviewer for highlighting these recent studies. We have cited and discussed these in the revised submission. See Lines 116-118 & Lines 311-313.

We have also added further journal references detailing some of the recent RSV outbreaks that have occurred in several countries during 2020 and 2021.

4. Line 194: "Despite the additional sequences, the G protein phylogeny found that the viruses from the 2020-2021 epidemics did not cluster with any other viruses sampled nationally or internationally up to date."

A comment on how reduced Australian and global sequencing might influence this analysis and the conclusions on the level of observed diversity.

Per reviewers' request, we have reflected on this on the manuscript. See Lines 268-279.

REVIEWERS' COMMENTS

Reviewer #1 (Remarks to the Author):

The authors have substantially revised their manuscript to include sophisticated molecular clock and positive-selection detection. I think the manuscript is substantially strengthened by the inclusion. I have no further concerns.

Reviewer #2 (Remarks to the Author):

I commend the authors who have done an excellent job of revising their manuscript and have carefully addressed all of my earlier concerns.